# Intensive Motion Style Acupuncture Treatment (MSAT) Is Effective for Patients with Acute Whiplash Injury: A Randomized Controlled Trial

**DOI:** 10.3390/jcm9072079

**Published:** 2020-07-02

**Authors:** Doori Kim, Kyoung-Sun Park, Jin-Ho Lee, Won-Hyung Ryu, Heeyoung Moon, Jiwon Park, Yong-Hyun Jeon, Ji-Yeon Seo, Young-Joo Moon, Jin Namgoong, Byung-Cheul Shin, In-Hyuk Ha

**Affiliations:** 1Bucheon Jaseng Hospital of Korean Medicine, Bucheon 420010, Korea; doori.k07@gmail.com (D.K.); rwh0225@naver.com (W.-H.R.); mistymoon91@naver.com (H.M.); park-9106@hanmail.net (J.P.); pezflux@hanmail.net (Y.-H.J.); wowpan21@gmail.com (J.-Y.S.); myj0617@jaseng.org (Y.-J.M.); bomagi123@jaseng.org (J.N.); 2Jaseng Spine and Joint Research Institute, Jaseng Medical Foundations, Gangnam-gu, Seoul 100011, Korea; lovepks0116@gmail.com; 3Jaseng Hospital of Korean Medicine, Gangnam-gu, Seoul 100011, Korea; jhlee3006@jaseng.co.kr; 4School of Korean Medicine, Pusan National University, Yangsan, Kyungnam 50612, Korea; 5Spine & Joint Center, Pusan National University Korean Medicine Hospital, Ysangsan, Kyungnam 50612, Korea

**Keywords:** neck pain, range of motion, whiplash injuries, Korean Traditional Medicine, acupuncture therapy

## Abstract

In this single-center, parallel, randomized controlled trial, we aim to examine the effects and safety of motion style acupuncture treatment (MSAT; a combination of acupuncture and Doin therapy) on pain reduction and functional improvement in patients with whiplash-associated disorders (WADs). Ninety-seven patients with cervical pain admitted to the Bucheon Jaseng Hospital of Korean Medicine, South Korea, due to acute whiplash injury were treated with integrative Korean medicine (IKM) with (MSAT group, 48 patients) or without (control group, 49 patients) an additional 3-day MSAT during hospitalization (5–14 days) and followed-up for 90 days. The mean numeric rating scale (NRS) scores of the MSAT and control groups at baseline were 5.67 (95% confidence interval (CI), 5.33, 6.01) and 5.44 (95% CI, 5.06, 5.82), respectively, and on day 5, 3.55 (95% CI, 3.04, 4.06) and 4.59 (95% CI, 4.10–5.07), respectively. The NRS change difference between the groups was −1.07 (95% CI, −1.76, −0.37). The rate of recovery of neck pain (NRS score change ≥ 2 points) was significantly faster in the MSAT than in the control group (log-rank test *p* = 0.0055). IKM treatment combined with MSAT may be effective in reducing the pain and improving the range of motion in patients with WADs.

## 1. Introduction

The term “whiplash-associated disorders” (WADs) was first introduced by the Quebec Task Force in 1995 and it refers to a wide variety of bony or soft tissue injuries that usually occur after a traffic accident [1]. WADs are a common cause of chronic neck pain in developed countries [2]. Among the typical symptoms are cervical pain and stiffness, and pain at the thoracic, shoulder, and temporomandibular joints, which could be accompanied by accessory symptoms, including vertigo, visual impairments, fatigue, sleep disorder, anxiety, depression, and psychological stress [3,4].

WAD is a prevalent health issue, affecting approximately 83% of patients involved in traffic accidents [2] and has economic effects as it causes a serious economic burden on the healthcare system [5,6]. In 2018, the economic cost due to the WADs was approximately GBP35.6 billion (USD46.2 billion) in the United Kingdom [7] and approximately USD3.9 billion and EUR10 billion in the United States and Europe, respectively [8]. Therefore, WADs have emerged as a major socioeconomic problem in South Korea and worldwide, highlighting the urgent need for a fast and effective intervention to reduce the casualties due to WADs.

Meanwhile, South Korea has an integrated health care system that combines western and Korean traditional medicine, which are both covered by traffic accident insurance. The number of patients seeking Korean traditional medicine after a traffic accident and the related expenses have increased. Indeed, Korean traditional medicine accounted for 27.7% of all traffic accident-related treatments in 2016 [9,10,11]. Acupuncture treatment is the main method of Korean traditional medicine, and the existing studies based on the effects of acupuncture treatment on cervical pain after whiplash injury have generally highlighted its effectiveness [12,13,14].

Additionally, Doin therapy is a common therapy used by Korean physicians to assess the abnormal or limited movements in patients and it utilizes a breathing method and passive or active movements for treatment [15,16]. Its efficiency in improving pulmonary function and in the treatment of a variety of symptoms and diseases, including chronic obstructive pulmonary disease, hepatitis B, and irritable bowel syndrome, has been reported [17].

Motion style acupuncture treatment (MSAT) is a method combining acupuncture treatment with Doin therapy [18]. In MSAT, the traditional acupuncture treatment is followed by passive or active movement of the patient’s body while the acupuncture needles are inserted. All movements are performed under the supervision and direction of a physician. In South Korea, MSAT is used to relieve musculoskeletal pain in clinical practice. Interestingly, in recent years, its use in South Korea and China has been increased as it enhances the effectiveness of traditional acupuncture treatment [19,20,21]. However, there is limited evidence regarding the efficacy and safety of MSAT, as very few studies have been conducted to date [22].

Therefore, we aim to examine the effect of MSAT combined with the integrative Korean medicine treatment (MSAT-IKM treatment) on pain reduction and functional improvement in patients with cervical pain after acute whiplash injury and to evaluate its safety in a randomized controlled trial (RCT). In a situation in Korea, where IKM is covered by traffic accident insurance and many traffic accident patients seek Korean medicine, IKM could be considered to be a general treatment for patients with traffic accidents in Korea. This study was conducted in accordance with the CONSORT 2010 and the STRICTA reporting guidelines.

## 2. Material and Methods

### 2.1. Study Design

This parallel, single-centered, pragmatic, randomized trial was conducted at the Bucheon Jaseng Hospital of Korean Medicine, South Korea, from July 2019 to December 2019. The participants were selected from July 2019 to September 2019 among hospitalized patients with WADs.

This study protocol received approval from the Institutional Review Boards of Jaseng Hospital of Korean Medicine (approval no., JASENG 2019-05-004) and was registered at ClinicalTrial.gov (NCT04113824; https://clinicaltrials.gov/ct2/show/NCT04113824?cond=MSAT&draw=2&rank=2). All participants were provided written informed consent before randomization and after being informed regarding the study design by a researcher.

### 2.2. Participants

#### 2.2.1. Inclusion Criteria

The inclusion criteria for the participants were aged between 19 and 70 years; the need for hospitalization due to acute whiplash injury that occurred within 7 days of a traffic accident; a pain numeric rating scale (NRS) score of 5 or higher at baseline; agreement to participate in this study by providing written informed consent.

#### 2.2.2. Exclusion Criteria

The exclusion criteria were (1) diagnosis of a certain serious disease that can cause back pain (e.g., malignant tumor, spinal infection, inflammatory spondylitis); (2) progressive neurologic deficits or serious neurologic symptoms; (3) history of neck surgery or treatment at 3 weeks prior to the study; (4) the cause of pain was a connective tissue disease rather than a spine disorder (e.g., tumor, fibromyalgia, rheumatic arthritis, and gout); (5) presence of a chronic disease that could limit the therapeutic effect or interfere with the interpretation of results (e.g., cardiovascular disease, kidney disease, diabetic neuropathy, dementia, and epilepsy); (6) steroid, immunosuppressant, psychiatric or other drug administration that could influence the results of the study; (7) inappropriate or unsafe performance of acupuncture treatment due to an infection risk (e.g., patients with hemorrhagic disease, anticoagulant treatment, severe diabetes, and severe cardiovascular disease); (8) pregnancy or planning pregnancy; (9) a serious mental illness; (10) participation in other clinical trials, except for observation studies without therapeutic intervention; (11) doubts regarding the written informed consent provision; (12) doubts for participating in the trial, as deemed by the researchers.

### 2.3. Randomization and Allocation Concealment

Randomization was conducted by a screening researcher. Eligible participants were assigned in a 1:1 ratio to the MSAT group or the control group. The MSAT group received IKM treatment with three sessions of MSAT, while the control group received IKM treatment only. Upon enrollment, each participant was assigned to one of the groups by a simple randomization method using a computer random generator [23,24]. The allocation concealment was achieved by using central allocation.

### 2.4. Blinding

As blinding of participants and physicians to the group assignment was impossible due to the nature of the study design, only single blinding was used for the assessing physicians. Researchers, who were blinded to the assignment of treatment groups and did not participate in the treatment, assessed the participants in a separate space prior to treatment.

### 2.5. Sample Size

The sample size was estimated using the mean difference of NRS for neck pain in the pilot study. Based on a pilot study conducted prior to this study (Appendix A), the effect size was estimated at 1.09. The difference in the mean NRS scores between the two groups was 1.44, and the pooled standard deviation (SD) of the change in the NRS scores from the two groups was 1.32.

To achieve 80% power at a significance level of 5%, it was calculated that a total of 28 participants (14 per group) was required. A conservative anticipated dropout rate of 30% was assumed, and considering the subgroup analysis, a total of 100 participants were enrolled (50 per group).

### 2.6. Interventions

All MSAT and IKM treatments were performed by Korean medicine doctors who had at least 3 years of clinical experience and were trained for standardized Korean medicine treatment and MSAT at the Bucheon Jaseng Hospital of Korean Medicine. All patients received IKM treatment during hospitalization, and those in the MSAT group received MSAT at hospitalization days 2, 3, and 4, three times in total.

#### 2.6.1. Control Group: Integrative Korean Medicine Treatment

IKM treatment consists of acupuncture, pharmacopuncture, chuna, and herbal medicine. In acupuncture treatment, the physician inserts the disposable acupuncture needles (30 × 0.25 mm; Dong-bang Acupuncture, SeongNam, South Korea) into the left and right trapezius muscles at the essential and optional acupuncture points at his/her discretion, with 6–12 points in total. The essential acupuncture points are SI15 (肩中兪, gyeonjungsu), TE15 (天髎, cheollyo), and LI16 (巨骨, geogol). According to the characteristics of each acupuncture point, perpendicular (直刺, chikcha) or oblique insertion (斜刺, saja) was used, followed by twisting and rotating of the needles for Deqi sensation. The needles were retained for 15 min, and the treatment was performed twice a day. Pharmacopuncture is a new treatment in Korean medicine that combines acupuncture and herbal medicine [25]. Pharmacopuncture involved the injection of a fixed herbal extraction dose using a disposable syringe. Depending on the patient’s condition, *Copridis rhizoma*, *Harpagophytum radix*, or *Jungsongouhyul* is used at the physician’s discretion. Pharmacopuncture is given with acupuncture treatment. Chuna manual therapy is a semi-standardized manual therapy used in the Korean traditional medicine, in which the doctor uses his/her hands, part of his/her body, or tools to stimulate the patients’ body structures. All physicians were trained by the standardized chuna education course. The procedure was performed among several methods that could be applied to the cervical spine at the discretion of the physicians (Appendix A). This procedure was performed once a day for 5–10 min before or after acupuncture treatment. For herbal medicine treatment, the patients took 75 mL of Samul-tang or Hwalhyul-jitong tang based decoctions containing medicinal herbs, effective in improving blood circulation (Hwalhyul) and reducing pain (Jitong), twice a day after meals. The basic ingredients of herbal medicine taken by patients are *Angelica gigas Nakai*, *Poria cocas*, *Paeonia lactiflora, Cnidium officinale,* and *Commiphora myrrha Engler*.

#### 2.6.2. Motion Style Acupuncture Treatment Group: Integrative Korean Medicine and Trapezius Motion Style Acupuncture treatments

The MSAT group received IKM treatment and three sessions of trapezius MSAT during hospitalization days 2, 3, and 4. For MSAT, the patients sat down on a chair or on the floor with their waist, back, and neck straight, the chin slightly pulled toward the body, and the eyes looking straight ahead. In this position, the physician inserted three disposable acupuncture needles (30 × 0.25 mm; Dong-bang Acupuncture) to a depth of 5–10 mm at the trapezius muscles of the patients. Then, the physician asked each patient to rotate his/her head to the left and right in turn to a maximum range of motion and assessed whether there was a difference compared to the normal range of motion and other movement abnormalities. Considering the side with the smaller range of motion as the affected side, the left and right rotational movements were performed, with the affected side’s range of motion as the basis. Then, the patients were asked to inhale and exhale when making movements toward and away from the midline of the body, respectively. The aforementioned movements were repeated for 8–10 times while the physician checked whether the head and neck movements were normal. When a problem was noted, the physician placed each hand on the posterior part of the neck and the face, respectively, to guide the normal movement. After completing the repetitive movements, the maximum range of motion was checked again, and when the normal range of motion could not be achieved, the movement was repeated. The movement was performed for approximately 10 min at each treatment, and the degree and frequency of movement were adjusted at the discretion of the physician. Five physicians who had received training for MSAT and had more than 3 years of clinical experience performed MSAT. Details of the MSAT procedure are shown in Appendix A.

### 2.7. Outcomes

The assessing doctors were blinded to group assignment and did not participate in the treatment. Sex, age, body weight, smoking, alcohol use, and medical history were included in the baseline assessment. Baseline measurements were performed at day 2 before treatment, and the primary endpoint was assessed at day 5, just 1 day after the completion of the three MSAT sessions. Following the single screening, the participants were assessed for the primary and secondary outcomes at days 2, 5, and 8, and at discharge. The NRS and the visual analog scale (VAS) scores, and the cervical range of motion (ROM) were measured at days 2–5. Measurements were performed twice (before and after treatment) at day 2 and once (before treatment) at days 3–5. The patients were followed-up by a phone call interview at 14 days and at 3 months after discharge (Appendix A).

#### 2.7.1. Primary Outcome

The primary outcome of our study was the difference in the neck pain NRS scores between the two groups at day 5. In the NRS method, the patients rated their current pain from 0 to 10, with 0 and 10 indicating no pain and the worst pain imaginable, respectively [26,27].

#### 2.7.2. Secondary Outcomes

Visual Analog Scale

The VAS is a measurement tool that documents the degree of pain that a patient feels on a 100-mm-long line, with the end corresponding to no pain and the other to the worst pain imaginable. The patient marks a point on the line that matches the pain he/she feels [26,28]. The neck and arm pain VAS scores were measured before and after the first treatment at day 2, before treatment at days 3–5, and at discharge.

Neck Disability Index

The neck disability index (NDI) is an instrument for assessing the disability in performing activities of daily living that may result from neck pain. It consists of 10 items, scored 0–5, for a total of 50 points. Higher scores indicate greater disability in performing activities of daily living [28,29]. NDI was measured at days 2, 5, and 8, and at discharge.

Cervical Range of Motion

Maximum active cervical ROMs that can be achieved with respect to neck flexion, extension, right rotation, left rotation, right lateral flexion, and left lateral flexion were measured before and after treatment at day 2, before treatment at day 3–5, and at discharge.

Quality of Life

In the present study, the Korean version of the 5-level EuroQol-5 Dimension (EQ-5D-5L) was used and to assess the participants’ quality of life. It is the most widely used for such measurements as it assesses multiple aspects of the responder’s health state and indirectly estimates the quality weights for each health state. The EQ-5D-5L consists of five items assessing the responder’s current health state (mobility, self-care, usual activities, pain, and anxiety/depression), and the responder answers on a 5-point Likert scale (1, no problems; 2, slight problems; 3, moderate problems; 4, severe problems; 5, unable to solve the problems) [30].

Patient Global Impression of Change

The patient’s global impression of change (PGIC) is a measure for assessing patients’ improvement on a 7-point scale, where the patients rate their functional improvement after treatment (1, very much improved; 2, much improved; 3, minimally improved; 4, no change; 5, minimally worse; 6, much worse; 7, very much worse). This measure was originally developed for a psychological assessment, but it is now used in a variety of other medical fields to assess the degree of pain relief [31].

Adverse Events

The presence of adverse events (AEs) was assessed at each visit and documented on case report forms. All AEs were documented regardless of the presence of a causal relationship with the given treatment. The AEs were classified, after an internal meeting of the researchers, according to Spilker et al. [29], as mild (did not impair the participant’s normal activities of daily living (ADLs), caused minimal discomfort, and required no additional treatment), moderate (significantly impaired the participant’s normal ADLs and may have required treatment, but they were resolved after treatment), or severe (severely impaired the participant’s normal ADLs, required intense treatment, and left sequelae). Concerning their relationship to the treatment, AEs were classified according to the World Health Organization-Uppsala Monitoring Center causality scale [32] as definitely related, probably related, possibly related, probably not related, definitely not related, or unknown. Classifications were made after three Korean medicine physicians reviewed each case and reached consensus.

### 2.8. Statistical Analysis

All efficacy and safety analyses were assessed in the modified intention-to-treat population, which included all participants who underwent allocation, except those who had never been treated (one in the MSAT group, two in the control group). The missing values were handled by multiple imputations using the Markov Chain Monte Carlo method. Twenty imputed datasets were generated, and covariates for imputation were treatment group, sex, age, and body mass index (BMI).

The continuous and categorical variable data are expressed as means ± SD and numbers (percentages), respectively. The outcome measures of the groups at each time point are presented with 95% confidence interval (CI). Analysis of covariance (ANCOVA) was conducted to compare the primary and secondary outcomes between the groups for each point. The covariates used were baseline outcome, age, sex, and BMI. For PGIC, age, sex, and BMI were used for covariates. The difference between the two groups by ANCOVA was reported with 95% CI and *p*-value.

For sensitivity analysis, datasets without imputed values were used to analyze the degree of change from the baseline. Mixed-effects repeated measures analysis was performed across all the time-points for the primary and secondary outcomes. The time variable was included as a categorical variable, and cross-term variable of treatment and time were included to analyze the effects of MSAT according to time. Cross-term variables, which mean differences in least-square mean changes between the two groups at each day, were presented with 95% CI and *p*-value. An unstructured covariance structure was used.

Survival analysis was conducted for the primary outcome. A neck pain NRS score change of 2 or greater from baseline, which is considered the minimal clinically important difference (MCID), was considered as an indication of recovery [33]. The survival probability of the MSAT and control groups is presented as a Kaplan–Meier curve, and the difference between the two groups was tested using the log-rank test. Additionally, the number of patients at risk at each time point is presented, and the Cox proportional-hazards model was used to estimate the hazard ratio. The proportional hazards assumption was tested using Schoenfeld’s global test.

Interim analysis was not to be performed or patient recruitment to be discontinued unless the principal investigator decided that there was an unacceptable risk of serious AEs in the groups. Statistical analyses were conducted using the SAS software version 9.4 (SAS Institution, Cary, NC, USA).

### 2.9. Data Availability

The datasets analyzed for the present study were collected directly from the patients and will not be publicly shared due to conditions of ethical approval. The datasets are available from the author upon reasonable request.

## 3. Results

### 3.1. Participants

The recruitment and follow-up of the study participants are shown in Figure 1. Between 3 July and 3 September 2019, 350 patients who complained of having cervical pain due to a traffic accident were screened. Of these, 250 patients were excluded (182 did not meet the inclusion criteria, 42 declined to participate, and 19 for other reasons).

The remaining 100 patients were randomly assigned to the MSAT or the control group (50 patients per group). Two and one patients from the MSAT and the control group, respectively, withdrew their consent before the start of the first intervention and were excluded. Therefore, 48 and 49 patients in the MSAT and the control group, respectively, received the intended treatment and were included in the intention-to-treat analysis.

Five patients in the MSAT group were discharged before the completion of the intervention and excluded. Therefore, 43 patients completed the procedure. Of these, 41 (two patients did not respond) and 38 (five patients did not respond or refused to respond) patients participated in the phone interview at the 2-week and the 3-month follow-up, respectively. Similarly, 6 patients in the control group were discharged before the completion of the intervention, and finally, 43 patients completed the procedure. Of these, 36 (seven patients did not respond) and 34 patients (nine patients did not respond or refused to respond) participated in the phone interview at the 2-week and 3-month follow-up after discharge, respectively (Figure 1).

### 3.2. Baseline Characteristics

The baseline characteristics of the participants are shown in Table 1. The proportion of females was 52.1% and 61.2% in the MSAT and the control groups, respectively. The mean age was 41.4 ± 12.6 and 42.6 ± 13.0 years in the MSAT and the control groups, respectively. The mean length of stay was 8.7 ± 3.8 and 8.4 ± 3.9 days in the MSAT and the control groups, respectively. Finally, the NRS was 5.7 ± 1.2 and 5.4 ± 1.3 in the MSAT and the control groups, respectively. No significant differences were observed between the two groups in all baseline characteristics (Table 1).

### 3.3. Outcome Comparison between the Two Groups

The outcome measures and their difference between the two groups at each time point are shown in Table 2. At day 5, at which the primary endpoint was assessed, the mean neck pain NRS score was significantly lower (*p* < 0.001) in the MSAT, 3.55 (95% CI, 3.04, 4.06), than in the control group, 4.59 (95% CI, 4.10, 5.07). At day 5, the mean arm NRS, neck VAS, and arm pain VAS scores, NDI and PGIC scores were lower in the MSAT than in the control group, and the ROMs were greater in the MSAT than in the control group in all directions.

In the MSAT and control groups, the mean neck NRS pain score decreased with time, from 5.67 (95% CI, 5.33, 6.01) and 5.44 (95% CI, 5.06, 5.82) at baseline to 1.41 (95% CI, 0.80, 2.01) and 1.05 (95% CI, 0.43, 1.67) at the follow-up day 90 in the MSAT and the control group, respectively. In addition to the neck pain NRS scores, the arm pain NRS and the neck and arm VAS scores also decreased, while the ROMs increased with time in both groups.

The difference in the mean neck pain NRS score between the two groups increased from −0.76 (95% CI, −1.15, 0.38) immediately after treatment to −1.07 (95% CI, −1.76, −0.37) at day 5 when all three MSAT sessions were completed. The difference gradually decreased thereafter to −0.34 (95% CI, −1.19, 0.51) at discharge, and to 0.23 (95% CI, −0.67, 1.14) at 90-day follow-up, at which the difference was no longer statistically significant. In addition to the neck, arm pain NRS and VAS scores, NDI, and ROMs also showed a similar trend, as they first increased and then decreased. All outcomes (neck and arm pain NRS scores, NDI, EQ-5D, and PGIC) measured at the follow-up day 90 were not statistically different between the groups (Table 2).

### 3.4. Outcome Changes in Each Group

The mean neck pain NRS score decreased by 2.34 (95%CI, 1.89, 2.79) and 1.86 (95%CI, 1.41, 2.31) in the MSAT and the control groups, respectively, from baseline to day 5, and the change in the mean NRS score was significant for both groups. In addition, the difference in change between the two groups was −1.05 (95% CI, −1.57, −0.53), which showed that the change in the MSAT group was greater. Moreover, the degree of improvement in neck pain VAS score and all ROMs (flexion, extension, right/left rotation, and right/left lateral flexion) were greater in the MSAT group. A significant improvement in the neck pain NRS score was noted immediately after the first treatment in the MSAT compared with the control group. The difference remained significant until day 5 and was no longer significant at discharge and follow-up. Significant differences in ROM between the two groups were maintained at discharge. No significant differences in NDI, EQ-5D, and arm pain were found between the two groups (Table 3, Figure 2).

### 3.5. Survival Analysis of Neck Pain NRS Score Reduction

Survival analysis showed that the recovery rate of neck pain (NRS score change of 2 or greater) was significantly faster in the MSAT than in the control group (log-rank test, *p* = 0.0055). In the MSAT group, 27 out of 48 patients did not achieve an NRS score change of 2 at day 5, and 20 of them did not achieve it at discharge. In the control group, 38 patients did not achieve an NRS score change of 2 at day 5, and 32 of them did not achieve it at discharge. At discharge, the hazard ratio (HR) of the MSAT group with respect to the control group was 1.94 (95% CI, 1.15, 3.27), showing that the HR for recovery was higher in the MSAT group (Figure 3). The proportional hazards assumption was satisfied (Schoenfeld’s global test, *p* = 0.368).

### 3.6. Adverse Events

AEs occurred in a total of 20 patients, 13 and 7 in the MSAT and the control group, respectively (Appendix A). Except for one case, in which antihistamine was provided for pruritus (moderate), the severity of all AEs was mild and did not require specific treatment. With respect to each item, 12 had diarrhea and soft stools, 3 had nausea and heartburn, 4 had skin-related symptoms, including pruritus and hives, and one had dizziness. Of these, one patient received antihistamines through injection, cream, and orally due to unexplained generalized pruritus and rash, which improved after 10 days. Moreover, 2 patients complained of having pruritus in the face and lower back, which were not parts of the acupuncture treatment, but improved within two days without any specific treatment. One of the other skin-related symptoms was vesicle formation following cupping, and the vesicles disappeared two days after performing a simple dressing treatment. In addition, diarrhea and gastrointestinal symptoms due to herbal medicine as part of the IKM treatment and consumption of inappropriate food were reported.

## 4. Discussion

In this study, trapezius MSAT-IKM treatment was found to be more effective in pain reduction and ROM improvement compared with IKM treatment alone in patients with acute neck pain after a traffic accident. Although both MSAT-IKM and IKM treatments were effective in reducing neck pain and improving ROM and function, MSAT treatment particularly contributed to fast pain reduction and ROM improvement.

In spinal pain, an NRS score change of 2 is generally accepted as the MCID [33]. In this study, at day 5, at which the primary endpoint was assessed, the mean NRS score of the MSAT group decreased by more than 2, whereas the mean NRS score reduction in the control group was less than 2. However, at 90-day follow-up, the NRS score decreased by more than 2 points in both groups. In other words, the treatment results over the long-term were not different between the two groups, but during the initial phase of treatment, the degree of improvement in the MSAT group tended to be greater. Moreover, the same tendency was observed in the analysis of data without imputation. It implies that MSAT can be expected to have an immediate effect on acute pain relief, but the effects do not appear to last long. In particular, the NRS score and the ROM were significantly improved immediately after performing the MSAT, highlighting its immediate therapeutic effect.

All AE cases were classified as mild in this study, except for one case. GI symptoms were the most common AE, and a few instances of skin-related symptoms and dizziness were also observed. There were more cases of AE in the MSAT group, and diarrhea accounted for most of the difference between the two groups. However, as MSAT is a motion-inducing treatment technique, diarrhea is not considered to be associated with MSAT. It was also difficult to find a direct association between MSAT and other skin-related symptoms, because the area of symptoms was different from that of MSAT. Therefore, although there were more AEs in the MSAT group, the relationship between MSAT and all AEs were considered as “probably not related”.

IKM could be considered as a general treatment for patients involved in traffic accidents in Korea for several reasons. First, patients with traffic accidents can receive a series of IKM that are all covered by traffic accident insurance. Second, Korean traditional medicine accounts for a high percent of all traffic accident-related treatments. Finally, many clinical trials in Korea for patients involved in traffic accidents are based on IKM [15,34,35,36]. As for the effect of IKM, clinical practice guidelines (CPGs) [37] for the use of Korean traditional medicine in the treatment of patients with traffic accident-related injuries suggested that herbal medicine, acupuncture, chuna manual therapy, and pharmacopuncture are recommended in patients with WAD grades 1 and 2 [9]. In addition, the effects of IKM on patients with WADs have been demonstrated in several previous studies [15,34,35,36].

There has been no study on the mechanisms for the immediate pain reduction and functional improvement of MSAT. Therefore, the underlying mechanisms can only be assumed. One possible explanation is pain-related fear. It is important to relieve pain-related fear as patients with fear have a poor prognosis and can develop chronic pain [38,39,40]. Treatments that gradually expose a fear can correct a patient’s incorrect perception [41]. This could be applied to WAD patients. One of the characteristics of WAD patients is that ROM is very limited. This is largely responsible for the fear for movement due to acute pain. When performing MSAT, the physician increases the ROM of the neck with the needles retained using manual therapy. During the treatment process, patients may have a reduced fear of movement and therefore make it easier for them move. Indeed, in the present study, a noteworthy finding was the change in ROM in the MSAT group. Another possible explanation is “*deqi*”. *Deqi* refers to the various subjective feelings of the patient during acupuncture treatment [42]. *Deqi* is important for the mechanism underlying acupuncture. *Deqi* can increase the effectiveness of acupuncture by promoting the secretion of various neurotransmitters such as cytokines, prostaglandins, and acetylcholine [43,44]. MSAT is a strong stimulus because it is a motion-inducing treatment with acupuncture needles. As the physicians increase patients’ movement with the acupuncture needles, patients may feel stronger stimulation. Therefore, it can be assumed that the feeling of *deqi* may be greater than that of general acupuncture treatment, causing a rapid treatment effect. However, deqi sensation between two groups was not compared in this trial, and further research is needed in this regard. 

The fast recovery rate of the MSAT group was more evident in the results of survival analysis. As an NRS score reduction of 2 or greater is considered as a recovery, the recovery rate of the MSAT group was faster, and the HR of recovery was higher in this group (1.94 (95% CI 1.15, 3.27)).

In recent years, the treatment method of using passive and active movements after inserting acupuncture needles has increasingly been used in South Korea and China [20,21]. Lin et al. [45] reported that patients with acute low back pain found greater pain relief and ROM improvement in the MSAT compared with the control groups, including loxoprofen, physical therapy, and conventional acupuncture. Shin et al. [22] showed that the effects of MSAT on pain reduction and functional improvement were greater than those of diclofenac injection in patients with severe acute low back pain. Luo et al. [46] conducted a clinical trial on patients with cervical spondylosis and showed that the MSAT group had a superior short-term treatment compared with the conventional acupuncture group. However, the related studies were limited, had a small sample size, or they lacked randomization. Therefore, there is a need to provide evidence for the efficacy and safety of MSAT.

To the best of our knowledge, this is the first RCT that has evaluated the efficacy and safety of performing trapezius MSAT, and its significance lies on the fact that it demonstrates the immediate functional improvement effect and safety of trapezius MSAT on acute pain. A wide range of outcome measures was collected from 100 hospitalized patients, who were also followed-up for 90 days after discharge to check symptoms and changes in status. Additionally, all procedures and evaluations were performed by trained Korean medicine physicians.

The fact that the patients and physicians were not blinded was a limitation of this study. Due to the nature of the treatment method, blinding of the treatment provided was impossible. Although the blind assessor evaluated the outcome to ensure objectivity and minimize bias, the possibility of a placebo effect cannot be ruled out. Nevertheless, the placebo effect would not have been significant because the control group also received treatment such as acupuncture and chuna manual therapy. 

In Korea, traffic accident insurance guarantees coverage for damages in the event of death or injury from a traffic accident, according to the Guarantee of Automobile Accident Compensation Act In addition, Korea has an integrated health care system that combines western and Korean traditional medicine, which are both covered by traffic accident insurance. Therefore, most patients seeking Korean traditional medicine treatment for traffic accidents receive IKM such as herbal medicine, acupuncture, pharmacopuncture, and chuna manual therapy. However, IKM may not be appropriate for serious fractures or injuries requiring surgery. This should be considered when interpreting the results. In this situation, through discussions with many experts participating in a national project to develop clinical practice guidelines, it was decided to conduct trials with a pragmatic, rather than experimental, design [47]. There were ethical issues and difficulties in recruiting patients for the more limited treatment. A pragmatic design is less ideal for demonstrating the effectiveness of treatment than an experimental design, but has the advantage of being able to give practical help for patients and policy-makers because it closely mirrors the real world [47,48,49]. Nevertheless, an experimental trial would be the best way to demonstrate the effectiveness of the evaluated treatment, and conducting research in countries without a traffic accident insurance system should be considered in the future.

Although it was impossible in the situation to use an instrument with confirmed validity for the ROM measurements, such as a goniometer [50,51], the measurements were performed by physicians who received sufficient training on ROM measurements. Therefore, it seems that there were no biases that could affect the study results. Due to the nature of the study, we were unable to control the length of the stay. However, as there was no significant difference in the length of stay between the two groups, it would not have affected the results. In this study, subjective variables such as NRS and NDI, which are widely used and validated, were mainly used. Numerous studies have examined the mechanisms underlying acupuncture [52,53,54]. However, for MSAT, it will be necessary to further verify its effectiveness through mechanistic studies using animal models and/or functional magnetic resonance imaging.

## 5. Conclusions

Trapezius MSAT combined with IKM treatment would be effective in fast pain reduction of acute pain and improving ROM in patients with WADs. More well-designed and multicenter RCTs should be conducted to provide additional evidence for the effects of MSAT in patients with WADs.

## Figures and Tables

**Figure 1 jcm-09-02079-f001:**
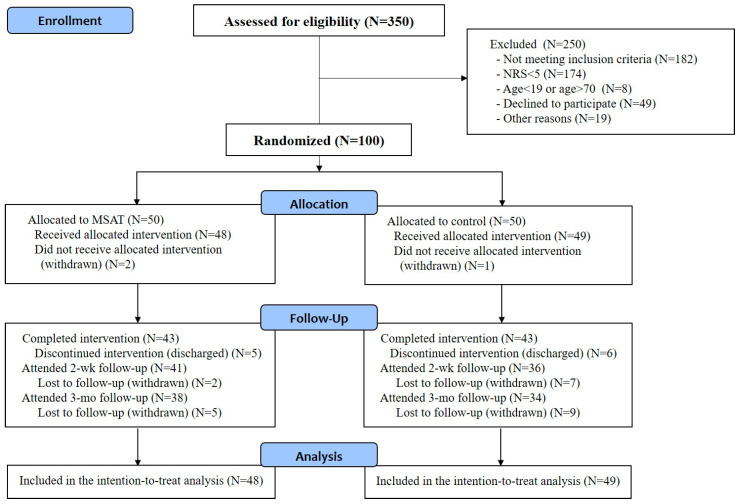
Flow chart of the participant enrollment process.

**Figure 2 jcm-09-02079-f002:**
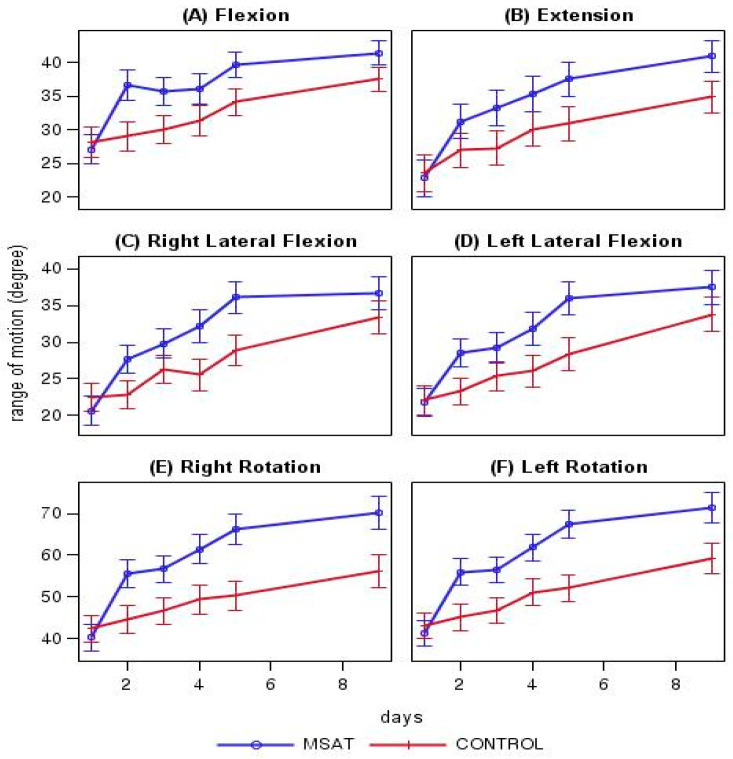
Comparison of range of motion between the MSAT and control groups. All figures show curves for the treatment period (baseline to discharge), with 95% confidence intervals represented by vertical bars. The plots and 95% confidence intervals were determined using a linear mixed model. From day 2 (after one session of treatment), the range of motion during all types of movements was greater in the motion style acupuncture treatment (MSAT) group than in the control group.

**Figure 3 jcm-09-02079-f003:**
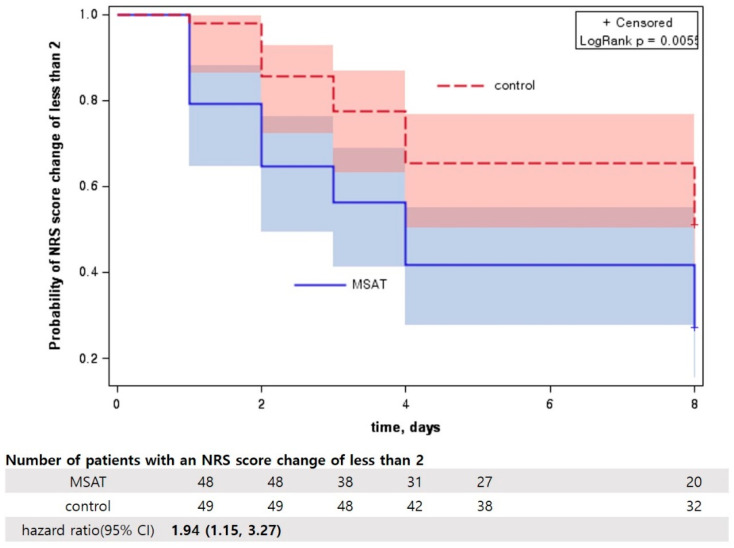
Kaplan–Meier curves for the probability of recovery in MSAT and control groups. Recovery (event) was defined as a change of more than 2 points in the NRS score for neck pain. The number of patients with an NRS score change of less than 2 is presented. The area filled with light blue and light red represents 95% CI. In the MSAT group, 27 of the 48 patients did not achieve an NRS score change of 2 on day 5, while 20 did not achieve the same at discharge. In the control group, 38 of the 49 patients did not achieve an NRS score change of 2 on day 5, while 32 did not achieve the same at discharge. The difference between the curves for the two groups was significant (log-rank test, *p* = 0.0055). The hazard ratio (95% confidence interval), calculated using the Cox proportional hazard model, was 1.94 (1.15, 3.27). NRS, numeric rating scale; MSAT, motion style acupuncture treatment.

**Table 1 jcm-09-02079-t001:** Baseline characteristics of patients with acute whiplash injury treated by treatment group.

		MSAT Group (*n* = 48)	Control Group (*n* = 49)
**Sex, *n* (%)**	Female	23 (52.1)	19 (61.2)
	Male	25 (47.9)	30 (38.8)
**Age (years)**		41.35 ± 12.62	42.63 ± 12.96
**Length of stay (days)**		8.73 ± 3.84	8.41 ± 3.91
**Height (cm)**		167.9 ± 8.97	167.92 ± 9.6
**Bodyweight (kg)**		68.73 ± 13.73	66.49 ± 12.12
**BMI (kg/m^2^)**		24.21 ± 3.34	23.44 ± 2.97
**Alcohol use, *n* (%)**	No	21 (43.7)	25 (51)
	Yes	27 (56.3)	24 (49)
**Smoking, *n* (%)**	No	29 (60.4)	30 (61.2)
	Yes	19 (39.6)	19 (38.8)
**NRS score for neck pain**		5.67 ± 1.17	5.44 ± 1.31

Notes: The values are presented as mean ± standard deviation or number (%). Abbreviations: BMI, body mass index; MSAT, motion style acupuncture treatment; NRS, numeric rating scale.

**Table 2 jcm-09-02079-t002:** Comparison of outcomes at each measuring point between the MSAT and control groups.

	Baseline	Day2	Day5	Discharge ^1^	90 Days f/u
**NRS score for neck pain**					
MSAT	5.67 (5.33, 6.01)	4.80 (4.39, 5.21)	3.55 (3.04, 4.06)	3.30 (2.78, 3.83)	1.41 (0.80, 2.01)
Control	5.44 (5.06, 5.82)	5.36 (4.98, 5.73)	4.59 (4.10, 5.07)	3.60 (2.95, 4.25)	1.05 (0.43, 1.67)
Difference ^2^	—	−0.76 (−1.15, −0.38)	−1.07 (−1.76, −0.37)	−0.34 (−1.19, 0.51)	0.23 (−0.67, 1.14)
*p*-value ^3^	—	0.0002	0.003	0.427	0.611
**NRS score for arm pain**					
MSAT	2.82 (2.04, 3.60)	2.51 (1.87, 3.15)	2.13 (1.55, 2.71)	1.39 (0.70, 2.09)	0.66 (0.05, 1.27)
Control	3.61 (2.91, 4.31)	3.40 (2.72, 4.08)	2.70 (2.03, 3.38)	2.49 (1.76, 3.23)	0.30 (−0.26, 0.86)
Difference ^2^	—	−0.23 (−0.60, 0.14)	0.01 (−0.66, 0.69)	−0.66 (−1.55, 0.22)	0.53 (−0.23, 1.28)
*p*-value ^3^	—	0.2247	0.971	0.142	0.174
**VAS score for neck pain**					
MSAT	5.74 (5.37, 6.12)	4.79 (4.31, 5.26)	3.64 (3.12, 4.17)	3.35 (2.78, 3.93)	—
Control	5.53 (5.13, 5.93)	5.31 (4.92, 5.71)	4.58 (4.09, 5.07)	3.56 (2.78, 4.33)	—
Difference ^2^	—	−0.66 (−1.13, −0.20)	−0.96 (−1.65, −0.26)	−0.34 (−1.19, 0.51)	—
*p*-value ^3^	—	0.0054	0.007	0.427	—
**VAS score for arm pain**					
MSAT	2.80 (2.02, 3.58)	2.49 (1.83, 3.16)	2.15 (1.55, 2.75)	1.46 (0.79, 2.13)	—
Control	3.62 (2.91, 4.34)	3.45 (2.76, 4.14)	2.84 (2.17, 3.50)	2.31 (1.54, 3.07)	—
Difference ^2^	—	−0.24 (−0.64, 0.17)	−0.05 (−0.71, 0.62)	−0.42 (−1.32, 0.48)	—
*p*-value ^3^	—	0.2501	0.890	0.359	—
**NDI**					
MSAT	40.94 (36.91, 44.97)	—	30.18 (25.87, 34.49)	27.89 (23.10, 32.68)	7.76 (3.90, 11.61)
Control	41.57 (37.94, 45.19)	—	32.61 (27.88, 37.33)	29.01 (23.83, 34.19)	4.26 (0.14, 8.37)
Difference ^2^	—	—	−2.21 (−8.22, 3.80)	−1.47 (−8.57, 5.62)	3.40 (−2.02, 8.83)
*p*-value ^3^	—	—	0.470	0.684	0.218
**ROM for flexion**					
MSAT	27.08 (24.26, 29.91)	36.67 (34.12, 39.21)	39.70 (37.14, 42.27)	41.22 (38.76, 43.68)	—
Control	28.16 (25.63, 30.70)	29.08 (26.37, 31.80)	34.34 (31.43, 37.26)	37.75 (35.28, 40.21)	—
Difference ^2^	—	8.50 (5.89, 11.11)	5.87 (2.09, 9.66)	3.66 (−0.01, 7.33)	—
*p*-value ^3^	—	<0.0001	0.002	0.050	—
**ROM for extension**					
MSAT	22.81 (19.24, 26.38)	31.25 (28.14, 34.36)	38.16 (34.92, 41.41)	40.81 (38.30, 43.32)	—
Control	23.57 (20.52, 26.62)	26.94 (23.92, 29.95)	30.68 (27.00, 34.37)	35.22 (31.89, 38.55)	—
Difference ^2^	—	4.64 (1.40, 7.89)	7.42 (2.42, 12.42)	5.40 (1.11, 9.69)	—
*p*-value ^3^	—	0.0055	0.004	0.014	—
**ROM for right rotation**					
MSAT	40.21 (36.12, 44.29)	55.63 (51.28, 59.97)	66.69 (61.86, 71.51)	70.57 (65.56, 75.59)	—
Control	42.45 (38.76, 46.14)	44.69 (40.82, 48.57)	51.28 (46.00, 56.56)	57.55 (51.27, 63.83)	—
Difference ^2^	—	12.42 (8.33, 16.52)	16.14 (8.71, 23.57)	13.29 (4.60, 21.98)	—
*p*-value ^3^	—	<0.0001	0.000	0.003	—
**ROM for left rotation**					
MSAT	41.25 (37.73, 44.77)	55.94 (51.86, 60.02)	67.62 (63.03, 72.21)	71.41 (66.34, 76.48)	—
Control	43.16 (39.34, 46.98)	45.10 (41.40, 48.80)	52.58 (47.99, 57.16)	59.87 (54.49, 65.25)	—
Difference ^2^	—	12.13 (8.38, 15.87)	15.94 (9.21, 22.68)	11.80 (4.13, 19.47)	—
*p*-value ^3^	—	<0.0001	0.000	0.003	—
**ROM for right lateral flexion**					
MSAT	20.63 (18.14, 23.11)	27.60 (25.27, 29.94)	36.20 (33.14, 39.27)	36.76 (33.27, 40.25)	—
Control	22.45 (20.14, 24.76)	22.86 (20.57, 25.15)	28.74 (25.81, 31.67)	33.54 (30.55, 36.54)	—
Difference ^2^	—	5.80 (3.50, 8.11)	7.49 (3.15, 11.82)	3.03 (−1.85, 7.92)	—
*p*-value ^3^	—	<0.0001	0.001	0.222	—
**ROM for left lateral flexion**					
MSAT	21.77 (19.24, 24.30)	28.54 (26.24, 30.84)	36.36 (33.25, 39.46)	37.59 (34.44, 40.75)	—
Control	22.04 (19.81, 24.27)	23.27 (21.12, 25.41)	28.37 (25.35, 31.39)	33.67 (30.47, 36.87)	—
Difference ^2^	—	5.16 (3.02, 7.30)	7.90 (3.52, 12.28)	3.63 (−1.08, 8.34)	—
*p*-value ^3^	—	<0.0001	0.000	0.130	—
**EQ5D**					
MSAT	0.67 (0.63, 0.72)	—	0.73 (0.69, 0.77)	0.75 (0.71, 0.78)	0.88 (0.85, 0.91)
Control	0.67 (0.63, 0.72)	—	0.73 (0.69, 0.77)	0.74 (0.70, 0.78)	0.88 (0.84, 0.92)
Difference ^2^	—	—	0.00 (−0.05, 0.05)	0.00 (−0.04, 0.05)	0.00 (−0.04, 0.05)
*p*-value ^3^	—	—	0.991	0.872	0.846
**PGIC**					
MSAT	—	—	2.72 (2.44, 3.00)	2.38 (2.11, 2.65)	1.49 (1.21, 1.77)
Control	—	—	3.23 (2.96, 3.50)	2.65 (2.34, 2.97)	1.14 (0.82, 1.47)
Difference ^2^	—	—	−0.48 (−0.87, −0.10)	−0.24 (−0.67, 0.20)	0.36 (−0.07, 0.79)
*p*-value ^3^	—	—	0.014	0.284	0.097

Notes: Outcomes were analyzed according to the intention-to-treat principle, and missing data were imputed with multiple imputations. The dashes indicate that outcome measurements were not administered. The outcome measurements at the 14-day follow-up have been excluded because they were similar to the discharge outcomes. The values are presented with 95% confidence interval. ^1^ The mean length of stay in the MSAT and control groups was 8.73 ± 3.84 and 8.41 ± 3.91 days, respectively. Five and six patients in the MSAT and control groups, respectively, were discharged before treatment completion. ^2^ The difference between MSAT and control groups. Differences are shown as the mean and 95% confidential interval. Analysis of covariance was performed to calculate the differences and *p*-values. The covariates included each baseline outcome, sex, age, and BMI. ^3^ (*) Statistically significant (*p* < 0.05). Abbreviations: MSAT, motion style acupuncture treatment; f/u, follow-up; NRS, numeric rating scale; VAS, visual analog scale; ROM, range of motion; NDI, neck disability index; EQ-5D, EuroQol-5 Dimension; PGIC, patient global impression of change.

**Table 3 jcm-09-02079-t003:** Comparison of changes in outcome from baseline at each measuring point between the MSAT and control groups.

	Day 2	Day 5	Discharge	90 Days f/u
**NRS score for neck pain**				
MSAT ^1^	0.86 (0.64, 1.09)	2.18 (1.76, 2.60)	2.34 (1.89, 2.79)	4.15 (3.73, 4.57)
Control ^1^	0.08 (−0.15, 0.31)	1.40 (0.98, 1.82)	1.86 (1.41, 2.31)	4.02 (3.59, 4.45)
Group*days ^2^	−0.56 (−1.01, −0.10)	−1.05 (−1.57, −0.53)	−0.25 (−0.86, 0.35)	0.10 (−0.42, 0.61)
*p*-value	0.047 *	0.001 *	0.4896	0.755
**NRS score for arm pain**				
MSAT ^1^	0.31 (0.07, 0.55)	0.82 (0.28, 1.36)	1.36 (0.83, 1.90)	1.91 (1.33, 2.49)
Control ^1^	0.21 (−0.02, 0.45)	0.52 (−0.03, 1.07)	1.10 (0.57, 1.63)	2.78 (2.18, 3.37)
Group*days ^2^	−0.89 (−1.66, −0.12)	−0.81 (−1.49, −0.12)	−1.05 (−1.74, −0.36)	0.07 (−0.49, 0.63)
*p*-value	0.059	0.054	0.0134 *	0.825
**VAS score for neck pain**				
MSAT ^1^	0.96 (0.68, 1.23)	2.03 (1.65, 2.41)	2.32 (1.82, 2.82)	—
Control ^1^	0.21 (−0.06, 0.49)	0.88 (0.50, 1.25)	1.93 (1.43, 2.43)	—
Group*days ^2^	−0.53 (−1.04, −0.02)	−0.94 (−1.49, −0.39)	−0.17 (−0.86, 0.51)	—
*p*-value	0.088	0.005 *	0.676	—
**VAS score for arm pain**				
MSAT ^1^	0.30 (0.05, 0.56)	0.65 (0.26, 1.05)	1.33 (0.81, 1.86)	—
Control ^1^	0.18 (−0.08, 0.43)	0.62 (0.23, 1.01)	1.22 (0.70, 1.75)	—
Group*days ^2^	−0.96 (−1.74, −0.17)	−0.86 (−1.56, −0.15)	−0.94 (−1.64, −0.24)	—
*p*-value	0.047 *	0.046 *	0.029 *	—
**NDI**				
MSAT ^1^	—	5.16 (3.73, 6.59)	6.17 (4.29, 8.04)	16.16 (14.24, 18.07)
Control ^1^	—	3.66 (2.24, 5.08)	7.57 (4.38, 10.76)	7.57 (4.38, 10.76)
Group*days ^2^	—	−1.72 (−3.92, 0.48)	−0.17 (−2.49, 2.14)	1.11 (−0.60, 2.82)
*p*-value	—	0.197	0.901	0.284
**ROM of flexion**				
MSAT ^1^	−9.58 (−11.21, −7.96)	−12.63 (−14.95, −10.31)	−14.35 (−16.74, −11.96)	—
Control ^1^	−0.92 (−2.53, 0.69)	−6.00 (−8.29, −3.70)	−9.37 (−11.74, −7.00)	—
Group*days ^2^	7.59 (4.51, 10.66)	5.55 (2.80, 8.31)	3.90 (1.39, 6.41)	—
*p*-value	0.000 *	0.001 *	0.011 *	—
**ROM of extension**				
MSAT^1^	−8.44 (−10.56, −6.32)	−14.77 (−17.84, −11.69)	−18.14 (−21.29, −14.98)	—
Control ^1^	−3.37 (−5.47, −1.27)	−7.41 (−10.44, −4.37)	−11.45 (−14.58, −8.32)	—
Group*days ^2^	4.31 (0.74, 7.89)	6.60 (2.97, 10.23)	5.93 (2.60, 9.26)	—
*p*-value	0.048 *	0.003 *	0.004 *	—
**ROM for right rotation**				
MSAT ^1^	−15.42 (−17.94, −12.90)	−26.07 (−29.61, −22.53)	−29.93 (−34.14, −25.73)	—
Control ^1^	−2.24 (−4.74,0.25)	−7.85 (−11.36, −4.35)	−13.77 (−17.95, −9.58)	—
Group*days ^2^	10.93 (6.13,15.73)	15.98 (10.91, 21.04)	13.93 (8.28, 19.57)	—
*p*-value	<0.001 *	<0.001 *	<0.001 *	—
**ROM for left rotation**				
MSAT ^1^	−14.69 (−16.92, −12.45)	−26.08 (−29.45, −22.70)	−30.16 (−34.10, −26.23)	—
Control ^1^	−1.94 (−4.15, 0.27)	−8.91 (−12.25, −5.56)	−15.97 (−19.89, −12.06)	—
Group*days ^2^	10.84 (6.29, 15.38)	15.26 (10.60, 19.91)	12.28 (7.13, 17.42)	—
*p*-value	<0.001 *	<0.001 *	<0.001 *	—
**ROM for right lateral flexion**				
MSAT ^1^	−6.98 (−8.44, −5.52)	−15.48 (−18.01, −12.96)	−16.06 (−18.65, −13.47)	—
Control ^1^	−0.41 (−1.85, 1.04)	−6.38 (−8.87, −3.89)	−11.03 (−13.58, −8.47)	—
Group*days ^2^	4.75 (2.04, 7.45)	7.28 (4.30, 10.26)	3.21 (−0.01, 6.43)	—
*p*-value	0.004 *	<0.001 *	0.101	—
**ROM for left lateral flexion**				
MSAT ^1^	−6.77 (−8.18, −5.36)	−14.20 (−16.72, −11.69)	−15.71 (−18.28, −13.14)	—
Control ^1^	−1.22 (−2.62, 0.17)	−6.36 (−8.84, −3.88)	−11.78 (−14.32, −9.25)	—
Group*days ^2^	5.28 (2.68, 7.87)	7.57 (4.38, 10.76)	3.66 (0.44, 6.88)	—
*p*-value	0.001 *	<0.001 *	0.062	—
**EQ-5D**				
MSAT ^1^	—	−0.06 (−0.09, −0.02)	−0.07 (−0.10, −0.04)	−0.20 (−0.24, −0.16)
Control ^1^	—	−0.05 (−0.09, −0.02)	7.57 (4.38, 10.76)	7.57 (4.38, 10.76)
Group *days ^2^	—	0.00 (−0.03, 0.04)	0.00 (−0.04, 0.04)	−0.01 (−0.04, 0.02)
*p*-value	—	0.841	0.925	0.553

Notes: The outcomes were analyzed using a linear mixed model according to the intention-to-treat principle. The dashes indicate that outcome measurements were not administered. The outcome measurements at the 14-day follow-up have been excluded because they were similar to the discharge outcomes. ^1^ mean (95% confidence interval). Least-square mean change from baseline on each day. ^2^ mean (95% confidence interval). Differences in least-square mean changes between the two groups on each day. (*) Statistically significant (*p* < 0.05).Abbreviations: MSAT, motion style acupuncture treatment; f/u, follow-up; NRS, numeric rating scale; VAS, visual analog scale; ROM, range of motion; NDI, neck disability index; EQ-5D, EuroQol-5 Dimension; PGIC, patient global impression of change.

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
