# Peer review of "Intensive Motion Style Acupuncture Treatment (MSAT) Is Effective for Patients with Acute Whiplash Injury: A Randomized Controlled Trial"

_jcm, 2020, doi:10.3390/jcm9072079_

Round 1
Reviewer 1 Report
General comments:
The authors should be congratulated for their effort to assess additional treatment strategies for whiplash-associated disorders (WAD).
Usually the effectiveness of a new approach is compared to a "standard of care". The standard of care for acute WAD, which includes management of pain, return for usual activity, and if not resolving after 1 week - manual and physical therapy and behavioral intervention was issued in 1995 by the Quebec Task Force on whiplash-associated disorders (1), and is supported by a more recent systematic review (2).
This study however compared effect of motion style acupuncture (MSAT) to integrative Korean medicine treatment (IKM) in acute whiplash injury. If IKM (which consists of acupuncture, pharmacopuncture, chuna and herbal medidicnes) is a standard of care for this disorder in Korea, the authors should state so, and if possible, provide evidence for its effectiveness.
In addition, editing would be advised so that a reader would not have to struggle understanding rationale and the results of the numerous tables, figures, and other data.
Specific comments:
Line 124. Sample size: Was the pooled variance of 1.29 based on the assumption that the SD of two groups was the same? Table 1 shows different values. Please address the decision to use pooled variance when mean values and their SDs had different values. Calculating the Table 1 values, the estimated sample size would be 40 (20+20). Not much of a difference, but data should be more precise. Table S1 was not provided, nor did Tables S2 and S3 later on.
Line 180. Did you mean "drinking" be alcohol consumption?
Line 402. Adverse effects (AE). Table S4 was not provided as well. In this study AE were documented "regardless of the presence of a causal relationship with the given treatment" (line 231). While this approach may be debatable (Allen EN et al), the list of AE in each treatment group was not provided. Keeping in mind that the control group (IKM) included herbal therapy, it could be responsible for some of the allergic reactions mentioned in the text e.g. pruritus and hives (line 407), as well as diarrhea and GI symptoms (line 413). Please provide the distribution of AE according to the treatment groups.
References
1. Spitzer WO et al, Scientific monograph of the Quebec Task Force on whiplash-associated disorders: redefining "whiplash" and its management. Spine 1995; 20: 1S-73S
- 2. Wingham T et al. The Effectiveness of Conservative Management for Acute Whiplash Associated Disorder (WAD) II: A Systematic Review and Meta-Analysis of Randomised Controlled Trials PLoS One. 2015; 10(7): e0133415.
- 3. Allen EN et al, Eliciting adverse effects data from participants in clinical trials. Cochrane Database Syst Rev 2018(1): MR000039.
Reviewer 2 Report
The authors have published some clinical trial or protocol about motion style acupuncture treatment (MSAT) on acute low back pain and chronic neck pain. In this study they examined the effects and safety of MSAT on acute whiplash injury and try viewing the mechanism for the pain reduction of MSAT. But the study design is not complete and the mechanism analysis is almost exactly zero. Additionally, some sentences are similarly with their previously publishes.
The control group in this study is integrative Korean medicine treatment such as acupuncture, pharmacopuncture and herbal medicine. Beside acupuncture treatment, Copridis rhizoma, Harpagophytum radix, or Jungsongouhyul was used to injected depending on the patient’s’ condition (line148-149) and the patients were asked to receive 75 ml of Samul-tang and Hwalhyul-jitong tang decoctions containing medicinal herbs, effective in improving blood circulation (Hwalhyul) and reducing pain (Jitong), twice a day after meal (line153-155). The treatments are very complex and could make different effects on the pain of acute whiplash injury. The scientific control group is should be acupuncture alone, and treatment with the herbal extraction or herbal medicine should be added as new treatment group. Furthermore, the MAST groups should include the MAST with acupuncture group, the MAST with herbal extraction or herbal medicine group and the MAST with integrative Korean medicine as described in the manuscript.
The qualities of images of all Figures are very low and especially Figure 1 is not possible for reading.
Reviewer 3 Report
The topic is very interesting since the incidence of neck pain and the cost associated with it are high. Also, the work has interest because it is very difficult to obtain relevant information and search for evidences regarding Korean medicine treatments published in English. So, I would like to refer that the proposed aims are relevant and justify the study.
However, I think that there are some aspects that need clarification:
- It would be important to provide a figure with the design of the treatments. Also, not all the patients in each group completed the treatments. This information is not indicated in the results.
- MSAT and IKE treatments should be explained more in detail in the text. It is said that pharmacopuncture was made “depending on patients’ condition …. and at the practioner discretion”. Also, the authors do not provide detailed information on how Chuna or herbal medicine were performed.
- How many physician/doctors were involved in measurements/treatments?
- Except from ROM, all the outcomes are subjective variables, based on patients´ opinions.
- Cervical range motion was measured by the physician without any instrument that allow validation. In these conditions we cannot exclude the possibility of a bias in the results. This is a very important aspect as measurable parameters are an important contribution to the credibility of the effectiveness of the treatment.
- The patients were randomised into two groups, one received MSAT treatment and the other did not. With this kind of study, it cannot be ruled out a possible placebo effect. This aspect deserves further discussion in the manuscript.
- It would be interesting to discuss the lasting effects after day 5, as at 90 days follow up there is no differences between both groups. In Figure 1 appears a 2-wk follow up, but no results are shown.
- Authors referred that all AEs were mild, except in one case, and that they were not related with MSAT treatment. Nevertheless, the number of AEs in the experimental groups was higher than in the control group. This should be more explored in the text.
- The limitations of the study could be more explored in the discussion and, taking into account these limitations, the conclusions should be more cautious.
- It would be of interest to be referred in the text how future work is going to be conducted to provide additional evidence of the effect of MAST in patients with WAD.
Round 2
Reviewer 1 Report
- The authors argue that standard of care for WAD in Korea is IKM, "which is accounted for 27.7% of all traffic –related treatments". They do not provide however the evidence for its effectiveness other than that it is covered by traffic accident insurance. If the purpose of this study is to provide data that could be generalized besides Korean medical system, this information is crucial.
- Scientific and language editing is still needed (very much).
- Thank you
- Thank you
- AE. The whole paragraph is not clear enough. Perhaps the authors may consider another definition of AE.
Reviewer 2 Report
The authors have added some viewing on the mechanism for the pain reduction of MSAT. But the mechanism analysis is necessary to be examined and be provided in this study for publication in the Journal of Clinical Medicine.
The scientific control group is should be acupuncture alone, and treatment with the herbal extraction or herbal medicine should be added as new treatment group. Furthermore, the MAST groups should include the MAST with acupuncture group, the MAST with herbal extraction or herbal medicine group and the MAST with integrative Korean medicine as described in the manuscript.
